# Projection of the health and economic impacts of Chronic kidney disease in the Chilean population

**Magdalena Walbaum**[1], **Shaun Scholes**[1], **Rubén Rojas**[2], **Jennifer S. Mindell**[1], **Elena Pizzo**[3]*

**1** Research Department of Epidemiology and Public Health, University College London, London, United Kingdom, **2** School of Health and Related Research, University of Sheffield, Sheffield, United Kingdom, **3** Department of Applied Health Research, University College London, London, United Kingdom

* e.pizzo@ucl.ac.uk

**Data Availability Statement:** The full data sets of the national health surveys (Encuesta Nacional de Salud ENS) can be accessed in through the Ministry of Health of Chile website found at: http://epi.minsal.cl/encuestas-poblacionales/.

## Abstract

### Background

Chronic Kidney Disease (CKD) is a leading public health problem, with substantial burden and economic implications for healthcare systems, mainly due to renal replacement treatment (RRT) for end-stage kidney disease (ESKD). The aim of this study is to develop a multistate predictive model to estimate the future burden of CKD in Chile, given the high and rising RRT rates, population ageing, and prevalence of comorbidities contributing to CKD.

### Methods

A dynamic stock and flow model was developed to simulate CKD progression in the Chilean population aged 40 years and older, up to the year 2041, adopting the perspective of the Chilean public healthcare system. The model included six states replicating progression of CKD, which was assumed in 1-year cycles and was categorised as slow, medium or fast progression, based on the underlying conditions. We simulated two different treatment scenarios. Only direct costs of treatment were included, and a 3% per year discount rate was applied after the first year. We calibrated the model based on international evidence; the exploration of uncertainty (95% credibility intervals) was undertaken with probabilistic sensitivity analysis.

### Results

By the year 2041, there is an expected increase in cases of CKD stages 3a to ESKD, ceteris paribus, from 442,265 (95% UI 441,808–442,722) in 2021 to 735,513 (734,455–736,570) individuals. Direct costs of CKD stages 3a to ESKD would rise from 322.4M GBP (321.7–323.1) in 2021 to 1,038.6M GBP (1,035.5–1,041.8) in 2041. A reduction in the progression rates of the disease by the inclusion of SGLT2 inhibitors and pre-dialysis treatment would decrease the number of individuals worsening to stages 5 and ESKD, thus reducing the total costs of CKD by 214.6M GBP in 2041 to 824.0M GBP (822.7–825.3).

**Funding:** MW is funded by the programme Becas Chile from Comisión Nacional de Investigación Científica y Tecnológica (CONICYT), Chile. The funding source had no role in the study. EP is funded by the National Institute for Health Research (NIHR) Applied Research Collaboration (ARC) North Thames London. The views expressed in this publication are those of the authors and not necessarily those of the NHS, the NIHR or the Department of Health and Social Care. The funding source had no role in the study.

**Competing interests:** The authors have declared that no competing interests exist.

## Conclusions

This model can be a useful tool for healthcare planning, with development of preventive or treatment plans to reduce and delay the progression of the disease and thus the anticipated increase in the healthcare costs of CKD.

## Introduction

Chronic Kidney Disease (CKD) is a major global public health problem [1,2], with substantial implications for quality of life and economic burden on healthcare systems [3,4]. CKD is defined as decreased kidney function shown by glomerular filtration rate (GFR) of <60 mL/min/1.73m$^2$ (based on measured serum creatinine values), markers of kidney damage (e.g. albuminuria as indicated by increased albumin-to-creatine ratio (ACR) >30mg/dL), or both, of at least three months duration, regardless of the underlying cause [1,5]. There is conflicting evidence about the increase in the prevalence of CKD in the general population worldwide [6–8]. In Chile, the prevalence of reduced kidney function shown by the estimated GFR (eGFR <60 mL/min/1.73m$^2$) is currently 5.7% in the population aged 40 years and older [9]. Although the prevalence has not increased significantly during the past 10 years, it is expected to increase due to the ageing of the Chilean population and the increase in the prevalence of conditions contributing to CKD such as diabetes mellitus, hypertension and obesity [10].

Among the most important health complications of CKD are the development of cardiovascular disease (CVD) and the progression along the natural history of CKD to end-stage kidney disease (ESKD) and therefore the need for renal replacement treatment (RRT) such as dialysis or renal transplant [11,12]. Although CKD patients are between five and ten times more likely to die prematurely than to progress to ESKD [1], largely attributable to death from CVD [1,10,13], the prevalence of ESKD in the population is increasing [2,4,14], especially in countries with increasing elderly populations and better access to healthcare [1]. In Chile, well-documented registries have shown a significant increase in the population with ESKD [15–19]. The most recent report showed that around 1,300 patients per million population (pmp) are being treated with haemodialysis [18,19].

The economic burden of CKD contributes significantly to overall healthcare expenditure. Although most studies estimate the economic burden of ESKD, the increase in healthcare resources used in earlier stages of CKD is also significant [20]. There is an increase in the utilization of emergency departments, hospitalization, inpatient visits, medical expenditure and pharmacy costs [2,20,21]. In the United Kingdom, it was estimated that the total annual cost of CKD in 2009–10 was 1.45 billion GBP (or 1.87 billion GBP today when adjusting for inflation), equivalent to 1.3% of all National Health Service spending that year [22,23], with more than half spent on RRT [24]. In the United States, the US Renal Data System (USRDS) 2018 report estimated that Medicare expenditure for beneficiaries with CKD and ESKD was 23% of the total Medicare fee-for-service (FFS) spending [25–27] and around 7% of all Medicare´s expenditures [28]. In Chile, adults with ESKD in RRT represent 0.15% of the population but account for more than 3% of the national healthcare expenditures, as such, CKD accounts for the highest proportion of healthcare spending in the country [29].

Projecting CKD (e.g. the progression of cases from the earlier CKD stages to ESKD, and hence need for RRT) and quantifying the magnitude of its burden in the long term is therefore essential for policy-making, healthcare planning and management [2,30,31]. Thus, developing a multistate predictive model, with transparent and justifiable assumptions, able to simulate

the long-term pathways of CKD [31,32] can provide useful information for more effective healthcare planning, resource allocation, and ultimately the delivery of the best possible health-care service to patients [30]. In this study, we develop a population model to project CKD in Chile and estimate the future economic and health consequences of CKD from the perspective of the Chilean public healthcare system.

## Methods

### Model construction specifications

A dynamic stock and flow model was developed to represent the natural history of CKD for the Chilean population aged 40 years and older with reduced kidney function from stage 3a to death (Fig 1) with a time horizon of 20 years, up to 2041, to provide sufficient results of future healthcare costs and cases for decision making [25]. The model included six mutually exclusive states replicating the progression of the disease: from CKD stage 3a through to stage 5, with or without diabetes mellitus, ESKD with need of RRT, and death. The CKD stages were defined based on the Kidney Disease: Improving Global Outcomes (KDIGO) [5] 2012 classification (S1 Fig): stage 3a: eGFR 45–59 ml/min/1.73m$^2$ with normal or increased albuminuria; stage 3b: eGFR 30–44 ml/min/1.73m$^2$ with normal or increased albuminuria; stage 4: eGFR 15–29 ml/min/1.73m$^2$ with normal or increased albuminuria, and stage 5: eGFR <15 ml/min/1.73m$^2$ with normal or increased albuminuria. We could not include CKD stages 1 or 2, defined as normal eGFR but with increased albuminuria [5], as this variable was not measured in the whole sample of ENS 2016–17, so we had to limit our analysis to stages 3a to ESKD. Therefore, we refer hereafter to CKD as signifying stages 3a to ESKD excluding stages 1 and 2. The full model can be found in S2 Fig.

The progression of CKD was assumed based on the development of the disease and the available evidence reporting progression across the different stages. As time moves forward, a proportion of individuals remains in the same health state, some progress to the next CKD stage, began RRT or die [30], depending on the decrease of eGFR and mortality rates. Transitions were assumed to go in only one direction (Fig 1), due to the low possibility of regression or remission of the disease, and without skipping CKD stages [25]. Transition to death was possible from any of the other health states [31], depending on the difference in mortality rates

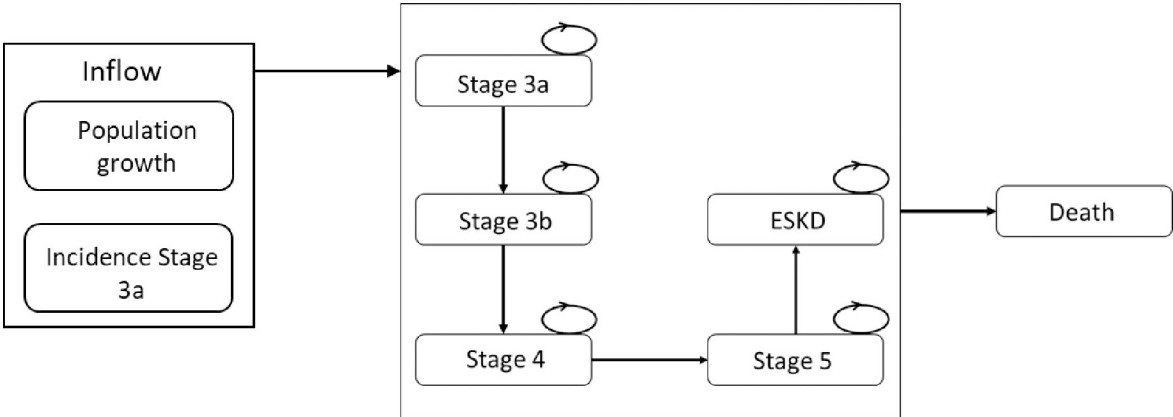

**Fig 1. Dynamic stock and flow model.** The inflow of the model was built based on the annual incidence of CKD stage 3a and the estimated growth in the Chilean adult population. The progression of CKD was determined by the estimated glomerular filtration rate (eGFR) levels calculated using the CKD-EPI equation. Progression could be decrements of 5, 3 or 1.4 ml/min/1.73 m$^2$ per cycle depending on the presence of diabetes and/or increased albuminuria. Patients can progress to another stage or remain in the same health state. Death could occur from any stage.

between CKD stages, age of the population and presence of comorbidities. Death was considered an 'absorbing' state [30].

The progression of CKD stages depended on the annual decrease of eGFR reported in the literature, with decrements of 1.4, 3 or 5 ml/min/1.73m$^2$ per year (depending on the presence of diabetes mellitus and/or increased albuminuria) [30,32–34]. The annual progression of eGFR assumed that individuals with CKD but with no diabetes or increased albuminuria would progress at a rate of 1.4 ml/min/1.73m$^2$ per year (slow progression); those with diabetes only would progress at a rate of 3 ml/min/1.73m$^2$ per year (medium progression); and those with diabetes and increased albuminuria in each stage would progress at a rate of 5 ml/min/1.73m$^2$ per year (fast progression) [30], replicating the proportion of individuals with diabetes and increased albuminuria in the Chilean population with CKD stages 3a to 5 (see Table 1). The proportion with hypertension was not included when determining the progression between stages, as 86% of the adults with CKD had survey-defined hypertension [35].

The model was developed from the perspective of the Chilean public healthcare system as CKD treatment is guaranteed for everyone diagnosed with CKD stages 3a to ESKD [36], independently of whether the individual belongs to the public or private healthcare sector. The model was limited to the Chilean population 40 years or older; it accounted for population-level heterogeneity in age, eGFR, albuminuria and diagnoses of hypertension and diabetes mellitus based on national data [9,18]. Ethnic differences were not included due to data limitations, and so the CKD-EPI equation used to determine eGFR assumed a white population [37].

## Data sources for the model parameters

Model parameters needed to populate the model were as follows: initial CKD prevalence; CKD incidence; population growth; prevalence of comorbidities; and mortality rates. These are set out in Table 1. Below we briefly discuss each in turn.

**Initial prevalence of CKD.** Prevalence data of CKD stages 3a-5, obtained from the nationally representative Chilean national health survey 2016–2017 (Encuesta Nacional de Salud, ENS 2016–17), was used for the initial distribution of eGFR and increased albuminuria in each stage and for the proportion of diabetes in the population with CKD. The sampling design and methods of data collection of the ENS 2016–17 have been reported elsewhere [9]. The estimated number of individuals at each discrete value of eGFR was obtained by fitting a linear regression with the natural logarithm of the number of cases as the outcome (y) and the discrete values of eGFR (range 1 to 59) as the independent variable (x). The fitted equation was $\ln(y) = -0.0556 + (0.077^*x)$; predicted values were obtained by using the exponential transformation (S3 Fig). For example, the estimated number at eGFR = 59ml/min/1.73m$^2$ was equal to $\exp(-0.0556 + (0.077^*59)) \approx 90$. By using the fitted equation of the discrete values of eGFR in the Chilean population, we could estimate the proportion of individuals in each stage that progressed based on the different progression rates assumed in the model.

In Chile, adults in need of RRT can receive kidney transplantation, haemodialysis (HD) or peritoneal dialysis (PD), although around 95% of Chilean patients in need of RRT are currently treated with HD and 4.5% with PD [16,18,19] with few patients receiving a transplant [16,18,38]. Thus, for this model, data about the ESKD group (initial prevalence, eGFR distribution, age, presence of comorbidities, mortality experience and costs) were obtained based on the information of patients having HD and PD from the Chilean Haemodialysis registries [16,18,19]. We therefore refer to RRT as signifying HD or PD excluding renal transplant.

**Incidence of CKD.** As one component of inflow, the annual incidence of CKD stage 3a among the general population and among those with diabetes was taken from the literature

**Table 1. Input values used for the stock and flow model.**

| Parameter | Mean value | 95% CI | Reference |
|---|---|---|---|
| *Prevalence of CKD in >40 years old* | | | |
| Stage 3a | 3.8% | 2.9–4.9 | ENS 2016–17 [9] |
| Stage 3b | 1.0% | 0.7–1.4 | ENS 2016–17 [9] |
| Stage 4 | 0.8% | 0.4–1.4 | ENS 2016–17 [9] |
| Stage 5 without RRT | 0.1% | 0.1–0.2 | ENS 2016–17 [9] |
| ESKD in HD or PD | 22310 | - | Haemodialysis Registries [18,19] |
| ESKD in PD | 1318 | - | Haemodialysis Registries [18,19] |
| *Proportion with increased albuminuria by Stage* | | | |
| Stage 3a | 35.5% | 25.7–46.8 | ENS 2016–17 [9] |
| Stage 3b | 44.4% | 30.4–59.5 | ENS 2016–17 [9] |
| Stage 4 | 77.0% | 47.0–92.7 | ENS 2016–17 [9] |
| Stage 5 | 100% | - | ENS 2016–17 [9] |
| *Incidence of CKD[a]* | | | |
| General adult population | 1330 pmp | 1253–1412 | Drey et al. [39] |
| Adults with DM | 2.2% to 4.3% | - | Koye et al. [40] |
| *Chilean adult growth* | 1.0% | | INE [41] |
| *Prevalence of comorbidities in CKD* | | | |
| Diabetes | 31.2% | 22.2–41.8 | ENS 2016–17 [9] |
| Hypertension | 86.2% | 78.8–91.3 | ENS 2016–17 [9] |
| *Annual progression of CKD* | | | |
| Fast progression[b] | 5ml/min | | Levy et al. [30], Hoerger et al. [32], Orlando et al. [34] and Wright et al.[33] |
| Medium progression[b] | 3ml/min | | Levy et al. [30], Hoerger et al. [32], Orlando et al. [34] and Wright et al.[33] |
| Slow progression[b] | 1.4 ml/min | | Levy et al. [30], Hoerger et al. [32], Orlando et al. [34] and Wright et al.[33] |
| *Mortality rates[c] (per 1000 pop)* | | | |
| 40–44 years old | 1.43 | | INE [41] |
| 45–49 years old | 2.2 | | INE [41] |
| 50–54 years old | 3.4 | | INE [41] |
| 55–59 years old | 5.4 | | INE [41] |
| 60–64 years old | 7.9 | | INE [41] |
| 65–69 years old | 12.2 | | INE [41] |
| 70–74 years old | 21.2 | | INE [41] |
| 75–79 years old | 33.6 | | INE [41] |
| 80–84 years old | 55.2 | | INE [41] |
| 85–89 years old | 94.5 | | INE [41] |
| 90–94 years old | 146.3 | | INE [41] |
| 95–99 years old | 228.2 | | INE [41] |
| 100+ years old | 260.6 | | INE [41] |
| *All-cause mortality HR per CKD Stage without DM[d]* | | | |
| Stage 3a | 1.19 | 1.10–1.29 | Fox et al. [42] |
| Stage 3b | 1.53 | 1.36–1.73 | Fox et al. [42] |
| Stage 4 | 2.27 | 1.86–2.77 | Fox et al. [42] |
| Stage 5 | 4.06 | 3.33–4.95 | Fox et al. [42] |
| *All-cause mortality HR per CKD Stage with DM[d]* | | | |
| Stage 3a | 1.18 | 1.07–1.30 | Fox et al. [42] |
| Stage 3b | 1.65 | 1.48–1.83 | Fox et al. [42] |
| Stage 4 | 2.28 | 1.91–2.72 | Fox et al. [42] |

(*Continued*)

**Table 1.** (Continued)

| Parameter | Mean value | 95% CI | Reference |
|---|---|---|---|
| Stage 5 | 4.46 | 3.26–6.10 | Fox et al. [42] |

CKD: Chronic kidney disease; DM: Diabetes mellitus; ENS: Encuesta Nacional de Salud (Chilean national health survey); HR: Hazard ratio; RRT: Renal replacement treatment. pmp: Per million population.

CKD stages (3a-ESKD) indicated by eGFR <60 mL/min/1.73 m$^2$ (determined by CKD-EPI equation) with normal or increased albuminuria in accordance with KDIGO guidelines [5].

[a] Incidence of CKD stage 3a, given in per million population (pmp).

[b] Fast, medium and slow progression considered as annual decline of eGFR of 5 mL/min/1.73m$^2$, 3 mL/min/1.73m$^2$ and 1.4 mL/min/1.73m$^2$ respectively.

[c] Mortality rates from general Chilean population in per thousand population.

[d] The reference category was eGFR 90–104 mL/min/1.73m$^2$.

[39,40] due to lack of Chilean data. The incidence of CKD stage 3a was estimated by the age-categories for the group age 40 years and older [39], it was adapted based on the age distribution of the Chilean population with all CKD stages from ENS 2016–17, and then transformed to proportion. For the projection of future incidence, we estimated the growth of the adult population (40 years or older) using data provided by the Chilean National Institute of Statistics (INE) [41]. Newly incident cases of adults with CKD stage 3a are assumed to enter the model as time progress. Incidence of other stages was considered only as the proportion of individuals that progressed from the previous stages, to avoid overestimation of new cases.

**Presence of comorbidities.** The prevalence of diabetes and hypertension per CKD stage in this population were estimated from the ENS 2016–17 data and the Chilean Haemodialysis registries (Table 1).

**Mortality rate.** All-cause mortality rate for the population 40 years and older was calculated using the Chilean data for all-cause mortality rate by age group provided by INE [41], and then adjusted with the hazard ratios of all-cause mortality per CKD stage found in the literature [42]. For adults with ESKD, the mortality rate was compared with the rate published in the Haemodialysis registries [16,18,19].

## Inclusion of sodium-glucose cotransporter-2 (SGLT2) inhibitors and pre-dialysis treatment

We simulated the effect of including the treatment with SGLT2 inhibitors for individuals with CKD stages 3a and 3b with diabetes mellitus [43,44]. Based on the available evidence, we assumed that this treatment decreased the progression of CKD (HR 0.71, 95% CI 0.57 to 0.89) [44,45]. The costs of treatment with SGLT2 inhibitors was provided by the Chilean Ministry of Health (calculated as £179.94 per patient annually).

Furthermore, we simulated the effect of including pre-dialysis treatment for individuals with CKD stages 4 and 5. Based on the evidence, we assumed that the inclusion of pre-dialysis treatment would reduce the progression of the disease (HR 0.85, 95% CI 0.74 to 0.98) [46]. The costs for the pre-dialysis treatment was estimated based on experts' opinion due to lack of Chilean data. The total annual cost per patient (in addition to the current treatment) was estimated at £360.97 and £1637.46 for stages 4 and 5, respectively.

## Costs

Only direct costs related to the treatment of the disease were included in the model. We incorporated these on an annual basis to represent the evolution of the expected total cost from the

perspective of the Chilean public healthcare system. The model was developed replicating the standard of CKD care in Chile. Currently, for stages 3a to 5, the Chilean Guidelines on CKD recommend multidisciplinary healthcare for patients [47]. All patients with stages 3a or 3b are usually diagnosed, managed and controlled in primary care by a multidisciplinary healthcare team and are referred to secondary care as the condition progresses to stage 4 and 5 [47]. For adults with ESKD, either HD or PD are provided, following the Chilean guidelines for this group of individuals [18,48–51].

The annual cost per stage was estimated, and subsequently multiplied by the number of expected cases per year, thus obtaining the total cost per year for the total CKD population. All data about resource consumption and their costs were obtained from the Individual Expected Cost Verification Study (EVC) [52] and the National Health Fund (FONASA) [53]. Constant rates were applied throughout the 20-year simulation period, therefore the variation of costs per year was given by the variation in the number of individuals estimated for each stage. All costs were adjusted by the variation of the consumer price index, with values as at June 2019: the adjustment was based on June 2019 values due to the economic instabilities that occurred during the end of 2019 and throughout 2020 in Chile due to the social crisis and the Covid-19 outbreak. This model was intended as a baseline to project the future cases and costs of CKD in Chile, and according to Chilean guidelines, a 3% annual discount rate was applied to the costs. Costs are presented in GBP using the UK HM Revenue & Customs May 2021 exchange rate [54]. The total annual costs for each stage and associated reference are outlined in Table 2. The full details of the treatments, frequency of use and costs considered for the model can be found in the S1–S5 Tables.

## Sensitivity analysis

Exploration of uncertainty in the modelled estimates was conducted with probabilistic sensitivity analyses (PSA). The PSA was undertaken whereby different probability distributions were associated with each parameter of the model depending on the data, using the mean value and the 95% confidence interval of the estimate [25]. For the costs data, we assumed a baseline cost ± 20% variation in the mean value to calculate the 95% confidence interval used in the PSA. The choice of the types of distributions was according to standard practice and to

**Table 2. Total annual costs per patient used for the stock and flow model.**

| Parameter | Mean value | Reference |
|---|---:|---|
| | **GBP £** | |
| Stage 3a | 48.76 | EVC [52] and FONASA [53] |
| Stage 3b | 48.76 | EVC [52] and FONASA [53] |
| Stage 4 | 66.73 | EVC [52] and FONASA [53] |
| Stage 5 without RRT | 990.12 | EVC [52] and FONASA [53] |
| ESKD with RRT | 12044.48 | EVC [52], FONASA [53] and Haemodialysis Registries [18,19] |
| Diabetes Mellitus | 51.37 | EVC [52] and FONASA [53] |
| SGLT2 inhibitors | 179.94 | Ministry of Health |
| Pre-dialysis stage 4 | 360.97 | Expert's opinion |
| Pre-dialysis stage 5 | 1637.46 | Expert's opinion |

GBP: British pound. EVC: Individual Expected Cost Verification Study. FONASA: Chilean National Health Fund. RRT: Renal replacement treatment. CKD stages (3a-ESKD) indicated by eGFR <60 mL/min/1.73m$^2$ (determined by CKD-EPI equation) with normal or increased albuminuria in accordance with KDIGO guidelines [5]. SGLT2 inhibitors: sodium-glucose cotransporter-2 inhibitors.

the nature of each parameter [55,56]. For probabilities, Beta distributions were used; for hazard ratios, normal distribution; and for costs, Gamma distribution. Monte Carlo simulations were carried out, with 10,000 iterations to reduce the Monte Carlo error across PSA [57].

We calibrated the model using international evidence and comparing the rates of increase in ESKD cases with the Chilean Haemodialysis registry [19]. All descriptive analyses to estimate the model parameters were adjusted for the complex survey design of the ENS and were performed using Stata V15.1 (StataCorp, College Station, Texas, USA). Stella Professional V 2.1 and Microsoft Excel Office 365 V2001 were used to construct the model and the different scenarios using the Visual Basic for Applications (VBA) macro fully parameterized to conduct the PSA. Figures were designed using Tableau Desktop Professional 2020.

## Results

The results of the model show that the number of Chilean adults with CKD, ceteris paribus, are projected to increase continually in all stages to the year 2041 (Table 3). The results from the PSA supported the continued growth of the CKD population in all stages as shown in Fig 2A.

### Baseline scenario

According to the baseline scenario, the number of individuals with CKD would increase from 442,265 (95% uncertainty interval) (441,808–442,722) in 2021 to 735,513 (734,455–736,570) in 2041. This represents an increase in the prevalence of CKD in the general population aged 40 years and older from the current 5.7% to an estimate of 7.5% in 2041. The number of people with ESKD and thus in need of RRT is projected to increase from 24,601 (24,590–24,613) in 2021 to 83,885 (83,682–84,088) in 2041.

In the baseline scenario, the costs of CKD are projected to increase from 322.4M GBP (321.7–323.1) in 2021 to 1,038.6M GBP (1,035.5–1,041.8) in 2041 (Table 4), with most of the costs due to the increase in RRT, from 287.3M GBP (286.7–287.8) in 2021 to 969.5M GBP (966.5–972.5) in 2041 (Fig 2B).

### Inclusion of SGLT2 inhibitors + pre-dialysis treatment

The results of including treatment with SGLT2 inhibitors for individuals with CKD stages 3a and 3b with diabetes and the introduction of pre-dialysis for individuals with CKD stages 4

**Table 3. Projection of the future number of Chilean adults with CKD by stage for years 2021–2041.**

| Year | Stage 3a (95%UI)* | Stage 3b (95%UI) | Stage 4 (95%UI) | Stage 5 (95%UI) | ESKD (95%UI) | Total (95%UI) |
|---|---|---|---|---|---|---|
| *Baseline scenario* | | | | | | |
| 2021 | 260,935 (260,641–261,228) | 105,803 (105,699–105,907) | 43,597 (43,555–43,638) | 7,329 (7,323–7,335) | 24,601 (24,590–24,613) | 442,265 (441,808–442,722) |
| 2026 | 280,848 (280,555–281,142) | 135,452 (135,277–135,626) | 59,588 (59,519–59,657) | 10,728 (10,715–10,741) | 33,082 (33,056–33,107) | 519,697 (519,122–520,273) |
| 2031 | 302,030 (301,729–302,330) | 155,641 (155,407–155,875) | 73,769 (73,649–73,890) | 14,215 (14,188–14,241) | 48,673 (48,607–48,740) | 594,328 (593,580–595,076) |
| 2036 | 326,919 (326,606–327,231) | 172,193 (171,922–172,463) | 84,950 (84,789–85,112) | 16,954 (16,916–16,992) | 66,615 (66,482–66,747) | 666,631 (666,715–668,546) |
| 2041 | 350,670 (350,347–350,994) | 187,330 (187,036–187,625) | 94,429 (94,239–94,619) | 19,198 (19,152–19,245) | 83,885 (83,682–84,088) | 735,513 (734,456–736,570) |
| *Inclusion of SGLT2 inhibitors + pre-dialysis treatment* | | | | | | |
| 2021 | 273,849 (273,615–274,082) | 100,157 (100,025–100,290) | 39,536 (39,460–39,612) | 6,032 (6,019–6,044) | 23,098 (23,071–23,125) | 442,672 (442,190–443,154) |
| 2026 | 305,886 (305,521–306,250) | 131,653 (131,552–131,754) | 50,446 (50,371–50,522) | 7,488 (7,473–7,503) | 26,025 (25,959–26,090) | 521,497 (520,876–522,118) |
| 2031 | 335,476 (335,025–335,927) | 156,805 (156,686–156,923) | 63,023 (62,962–63,084) | 9,775 (9,762–9,788) | 34,062 (33,962–34,161) | 599,141 (598,398–599,884) |
| 2036 | 366,746 (366,230–367,263) | 177,971 (177,815–178,127) | 74,430 (74,384–74,477) | 11,959 (11,948–11,969) | 45,471 (45,350–45,593) | 676,577 (675,726–677,428) |
| 2041 | 395,725 (395,159–396,290) | 196,872 (196,683–197,061) | 84,480 (84,440–84,519) | 13,877 (13,869–13,886) | 58,006 (57,872–58,140) | 748,960 (748,023–749,896) |

*UI: Uncertainty interval.

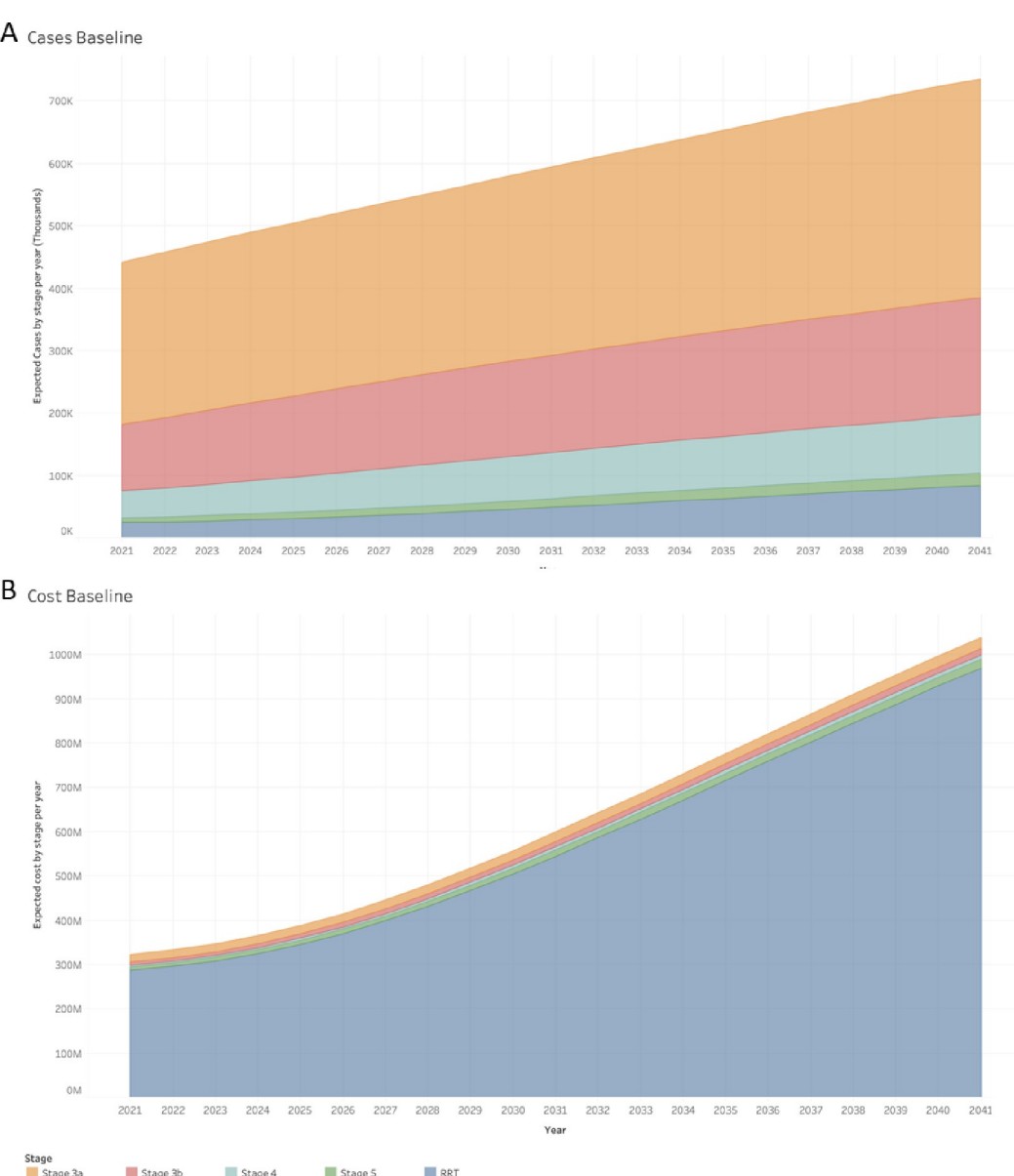

**Fig 2. Expected number of cases and costs per year and CKD stage.** Estimation of cases and costs of CKD to 2041. A: Estimation of cases of CKD by stage and year to 2041. B: Estimation of costs (in Million GBP) of CKD by stage and year to 2041.

and 5 are shown in Tables 3 (number of cases) and 4 (costs). With the inclusion of these treatments, the difference in the total number of cases of CKD by 2041 would be an increase of 13,447 (from 735,513 (734,455–736,570)) in the baseline scenario to 748,960 (748,023–749,896)), but with a marked change in the distribution of cases between CKD stages. Due to a higher percentage of people progressing at a slower rate, the number of cases in earlier stages (3a and 3b) at the end of 2041 would be higher than the baseline scenario and the total cases of stages 5 and ESKD would be fewer (Fig 3). This would reduce the total costs of CKD by 214.6M GBP projected for 2041 (from the 1,038.6M GBP in the baseline scenario to 824.0M GBP in the scenario with the inclusion of treatment with SGLT2 inhibitors and pre-dialysis) (Fig 4).

**Table 4. Projection of the future costs (in million GBP) of CKD by stage for years 2021–2041.**

| Year | Stage 3a (95%UI)* | Stage 3b (95%UI) | Stage 4 (95%UI) | Stage 5 (95%UI) | ESKD (95%UI) | Total (95%UI) | Discounted |
|------|-------------------|------------------|-----------------|-----------------|--------------|---------------|------------|
| *Baseline scenario* | | | | | | | |
| 2021 | 17.5 (17.5–17.6) | 7.0 (6.9–7.0) | 3.6 (3.5–3.6) | 7.1 (7.0–7.1) | 287.3 (286.7–287.8) | 322.4 (321.7–323.1) | 322.4 |
| 2026 | 19.8 (19.7–19.8) | 9.9 (9.9–9.9) | 5.4 (5.4–5.4) | 10.2 (10.1–10.2) | 369.4 (368.7–370.2) | 414.7 (413.8–415.6) | 357.7 |
| 2031 | 21.6 (21.5–21.7) | 11.9 (11.8–11.9) | 7.1 (7.0–7.1) | 14.0 (13.9–14.0) | 544.3 (543.1–545.6) | 598.9 (597.5–600.3) | 445.6 |
| 2036 | 23.7 (23.6–23.7) | 13.4 (13.3–13.4) | 8.4 (8.3–8.4) | 17.0 (16.9–17.0) | 758.6 (756.6–760.7) | 821.0 (818.8–823.3) | 527.0 |
| 2041 | 25.5 (25.5–25.6) | 14.7 (14.7–14.8) | 9.5 (9.4–9.5) | 19.4 (19.3–19.4) | 969.5 (966.5–972.5) | 1,038.6 (1,035.5–1,041.8) | 575.1 |
| *Inclusion of SGLT2 inhibitors + pre-dialysis treatment* | | | | | | | |
| 2021 | 38.4 (38.2–38.5) | 13.2 (13.2–13.2) | 14.5 (14.4–14.5) | 10.0 (10.0–10.0) | 276.1 (275.7–276.5) | 352.1 (351.5–352.7) | 352.1 |
| 2026 | 48.9 (48.7–49.1) | 21.1 (21.0–21.2) | 18.5 (18.4–18.5) | 11.8 (11.7–11.8) | 298.9 (298.5–299.2) | 399.1 (398.4–400.0) | 344.3 |
| 2031 | 56.6 (56.3–56.9) | 27.7 (27.6–27.8) | 23.6 (23.5–23.6) | 15.6 (15.5–15.6) | 382.0 (381.2–382.6) | 505.5 (504.4–506.5) | 376.1 |
| 2036 | 63.5 (63.1–63.8) | 33.2 (33.0–33.3) | 28.3 (28.3–28.4) | 19.3 (19.3–19.4) | 512.4 (511.7–513.1) | 656.7 (655.5–657.9) | 421.5 |
| 2041 | 69.5 (69.1–69.8) | 37.8 (37.6–37.9) | 32.5 (32.5–32.6) | 22.6 (22.6–22.7) | 661.7 (661.0–662.3) | 824.0 (822.7–825.3) | 456.3 |

*UI: Uncertainty interval.

## Discussion

To the best of our knowledge, this is the first population model to project CKD in the Chilean population and thus simulate the future CKD economic burden for the Chilean healthcare system. Using a dynamic stock and flow model, we represented the natural evolution of CKD in the Chilean population aged 40 years or older between 2021 and 2041. According to our

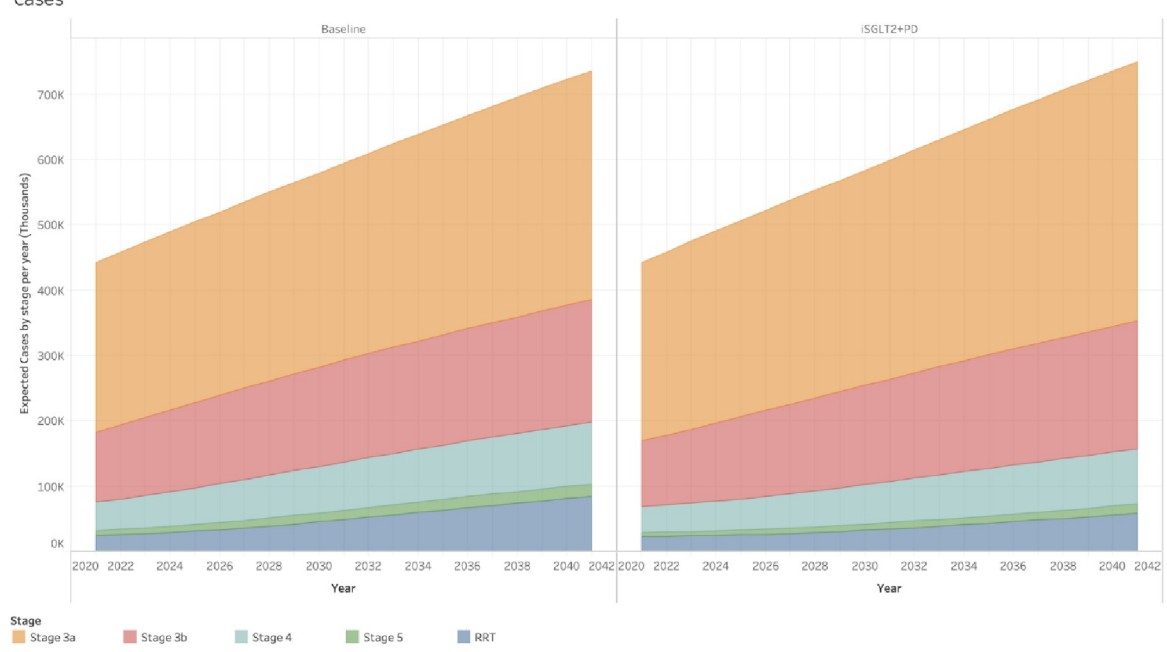

**Fig 3. Expected number of cases per year, CKD stage and scenario.** Comparison of expected number of cases between the two scenarios included in the study. The difference in the proportion of fast, medium and slow progressors between stages creates a difference in the distribution of individuals between them. In the baseline scenario individuals progress faster, thus the number of individuals in RRT by the end of 2041 is higher. Conversely, in the scenario that includes treatment with SGLT2 inhibitors for individuals with CKD stages 3a and 3b with diabetes mellitus, and pre-dialysis treatment for individuals with CKD stage 4 and 5, there would be a lower number of individuals in RRT by the end of 2041.

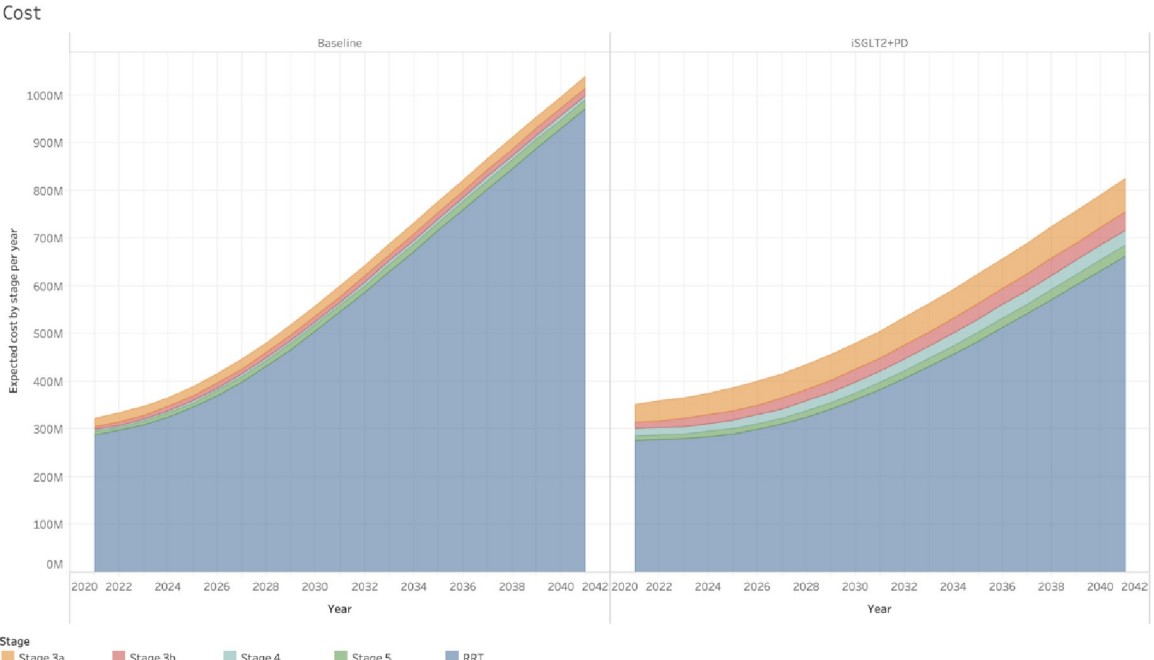

**Fig 4. Expected costs per year, CKD stage and scenario.** Comparison of expected costs (in Million GBP) between the two scenarios simulated in the study. The difference in the proportion of fast, medium and slow progressors between stages creates a difference in the distribution of individuals between them and therefore the healthcare costs. In the baseline scenario, with a greater number of individuals progressing faster to RRT, the total costs of CKD projected to 2041 is higher compared to the scenario with the inclusion of SGLT2 inhibitors for individuals with CKD stages 3a and 3b and with diabetes mellitus, and pre-dialysis treatment for individuals with CKD stages 4 and 5.

results, the projected number of Chilean adults with CKD stages 3a to ESKD would almost double in number in the next 20 years. This substantial increase in the population with CKD will increase healthcare resource utilization and overall costs accordingly [32]. Even though we included only direct costs of the treatment of CKD, our projection shows that they would triple by 2041. There are other important indirect costs such as hospitalizations and costs related to adverse events due to CKD that were not considered and these may have an important impact on the overall costs of the disease.

Studies projecting the disease in other countries have found similar results, with an increase in the prevalence of CKD and ESKD in the general adult population [2,28,32]. The study by Hoerger et al. [32] projected an increase in the prevalence of CKD stages 3a to ESKD in the United States from 13.2% in 2010 to 16.7% in 2030. Chile has a younger population structure compared with the United States; therefore, we would expect this difference in CKD prevalence, but the increase in percentage points has a similar trend between both countries. Wong et al. concluded that by 2050, nearly 25% of the population in Singapore would have CKD [2]. This significant difference with our projections is probably due to the inclusion of CKD stages 1 and 2 in their model (i.e. normal eGFR but with increased albuminuria). ENS 2016–17 measured albuminuria only in individuals with diabetes and hypertension; thus we could not classify adults at CKD stages 1 and 2 for the model. Additionally, we did not have sufficient information about the trajectory of these earlier stages in terms of the natural course and progression of the disease to be confident about including them in the model, therefore we limited our projection only to the population with reduced eGFR (CKD stages 3a-ESKD) with normal or increased albuminuria and considered the proportion of individuals with increased albuminuria to determine different progression rates. Nevertheless, we emphasize the importance

of considering all CKD stages to project the disease when national cohorts with long follow-up become available, to provide more reliable characterization of the relationship and progression of CKD and thus a more precise projection of it.

In our model we focus on CKD in adults aged 40 years or older because eGFR is relatively constant before that age [32]; eGFR begins declining around the age of 30–35 years so we estimated that relatively few cases were missed and this reduced the misclassification of cases in younger adults [6]. The risk of developing CKD and ESKD increases strongly with age [28,58], as it is related in part with the natural decline of kidney function, so the clinical significance of early stages of the disease (stages 3a-3b) has been debated for the more elderly population [6,59]. Nonetheless, we included this population in the model as the further decline in kidney function in this group resulting in progression to more advanced stages would lead to the need for more specialized medical treatment to prevent progression, or RRT in case they progress to ESKD, thus increasing healthcare resources utilization [42]. The overall Chilean adult population is projected to increase [41], leading to an expected increase in the burden of CKD in the following years [28].

Other risk factors behind the rise in CKD prevalence are probably the increase in the prevalence of diabetes, hypertension, and obesity in the general population. In Chile, according to the most recent health survey (2016–17), 12.3% of adults had survey-defined diabetes, 34.6% were obese and 27.7% had survey-defined hypertension [35]. These show a significant increase in diabetes and obesity compared with the 2009–10 survey, with prevalences of 9.4% and 25.7% respectively. These data support our projection of an expected increase in the number of individuals with CKD and emphasize the need to develop new interventions to slow the onset and progression of CKD at earlier stages and therefore reduce the incidence of ESKD [58]. As the results of the model simulations show, a reduction in the progression of the disease given by the estimated effect of the treatments: (1) SGLT2 inhibitors for those with diabetes mellitus in CKD stages 3a and 3b, and (2) pre-dialysis for those in CKD stages 4 and 5, would have a significant impact on the number of individuals advancing to later stages of the disease and therefore potential savings of around 214.6M GBP on direct costs for the Chilean healthcare system. These findings show the importance of targeting the progression of CKD as one of the key variables when establishing the treatment's goals. The evidence shows several options for effective treatments for the management of early stages of CKD to reduce progression of the disease such as the treatment with SGTL2 inhibitors for earlier stages of CKD [60–63] and the multidisciplinary pre-dialysis treatment for stages 4 and 5 [25,46], or conservative management for certain individuals with ESKD as an option instead of RRT [28,64]. These strategies need to be studied in future models with real world data to assess their implementation, cost-effectiveness and the impact on CKD in Chile.

Our results have several limitations. First, due to unavailability of Chilean cohort studies of CKD, the data sources used for the model may have introduced bias to our results. The cross-sectional design of the ENS 2016–17 and the estimate of eGFR based on single-point-in-time measurements of serum creatinine, are possible sources of bias. Moreover, due to the limitations of the cross-sectional design of the Chilean data we did not include important risk factors for development and progression of CKD in Chile such as obesity [28,65], different ethnicities [66], socio-economic level [67], other clinical risk factors [68], proteinuria [2], or acute kidney injury [69,70]. Likewise, adverse events of CKD such as myocardial infarction, stroke or other cardiovascular diseases [71], although important comorbidities for CKD [39,71], were not included in this first population model for CKD stages 3a to 5. These complications will have an important impact in the number of cases and the costs of the disease; therefore, they should also be considered in a future modelling study when longitudinal data becomes available. Also,

due to unavailability of Chilean renal transplantation data we could not include this treatment in the model and limited our simulation and analysis of RRT only to dialysis.

Secondly, our model assumed two simplifications regarding the progression and trajectory of eGFR. We permitted the progression of the disease only by decrease in eGFR in time, without allowing possible regression of eGFR. Nevertheless, we considered that the influence would be negligible as the evidence shows that only in a minority of patients does eGFR improve and often they revert to CKD [25]. The trajectory of eGFR was assumed to decrease linearly at three constant rates with difference between stages depending on the presence of diabetes mellitus and/or increased albuminuria [30]. These simplifications may have introduced some uncertainty to our results, that we tried to overcome with the PSA, but robust longitudinal data sets with longer follow ups are needed to have a deeper knowledge of both the progression and trajectory of eGFR between stages.

Thirdly, our estimates are based on current risk factor prevalence, CKD incidence, and all-mortality rates. If these rates change over time, with the inclusion of different interventions or with acute events such as the recent pandemic outbreak of Coronavirus (Covid-19), our estimates may either over- or under-estimate the CKD projections [32]. Therefore, these estimates must be treated with caution and must be validated when longitudinal data become available [30].

Our model provides essential information needed by decision-makers for future public healthcare planning; preparing resources needed; and taking effective actions to combat the problem of CKD in Chile. These results, although limited to the healthcare system perspective, indicate that the number of cases and costs of CKD would continue to increase in the future if no actions were taken. It is important to consider these results as part of a broader societal perspective, where CKD imposes large health and economic burden to individuals with the disease and their families, the national healthcare system and society (including the productivity loss of due to sick leave or early retirement). CKD and its risks factors such as diabetes and hypertension can be prevented or delayed, and therefore implementing effective prevention strategies to slow the increasing burden of the disease is an urgent public health priority.

## Supporting information

**S1 Fig. Classification of CKD.** *Source*: Kidney Disease: Improving Global Outcomes group. The numbers in the grid are the recommendations by KDIGO to the frequency of monitoring (in times per year).
(PDF)

**S2 Fig. Dynamic stock and flow model.** Dynamic Stock and flow model by Stella Professional V2.1.
(PDF)

**S3 Fig. Distribution of eGFR in Chilean national health surveys (ENS 2009–10 and 2016–17).** The estimated glomerular filtration rate (eGFR) levels were calculated using the CKD-EPI equation based on the data of the two most recent Chilean national health surveys (ENS 2009–10 and 2016–17). The distribution was assessed for eGFR <60 ml/min/1.73 m$^2$ with normal or increased albuminuria to estimate the proportion of individuals in each stage that would progress in one cycle to the next stage.
(PDF)

**S1 Table. Treatment included for CKD stages 3a and 3b.** Adapted from data extracted from the Chilean Individual Expected Cost Verification Study (EVC), the Chilean National Health Fund and experts' opinion. [a] Annual use per patient. [b] Frequency of use considered as the

percentage of patients that would use the specific treatment. [c] For simplification, we grouped all the laboratory tests considered for individuals in stages 3a and 3b. [d] Statins included: atorvastatin, lovastatin and pravastatin. [e] ACE inhibitors: Angiotensin-converting enzyme (ACE) inhibitors. Enalapril and captopril are included. [f] ARBs: Angiotensin receptor blockers (ARBs) or angiotensin II receptor antagonists. Losartan potassium is included.
(PDF)

**S2 Table. Treatment for CKD stage 4.** Adapted from data extracted from the Chilean Individual Expected Cost Verification Study (EVC), the Chilean National Health Fund and experts' opinion. [a] Annual use per patient. [b] Frequency of use considered as the percentage of patients that would use the specific treatment. [c] For simplification, we grouped all the laboratory tests considered for individuals in stage 4. [d] ACE inhibitors: Angiotensin-converting enzyme (ACE) inhibitors. Enalapril and captopril are considered. [e] ARBs: Angiotensin receptor blockers (ARBs) or angiotensin II receptor antagonists. Losartan potassium is considered. [f] Intravenous iron is considered for 30% of patients with stage 4, one or two times a week. [g] Erythropoietin is considered for 30% of patients with stage 4 two times a week.
(PDF)

**S3 Table. Treatment for CKD stage 5.** Adapted from data extracted from the Chilean Individual Expected Cost Verification Study (EVC), the Chilean National Health Fund and experts' opinion. [a] Annual use per patient. [b] Frequency of use considered as the percentage of patients that would use the specific treatment. [c] For simplification, we grouped all the laboratory tests considered for individuals in stage 5. [d] ACE inhibitors: Angiotensin-converting enzyme (ACE) inhibitors. Enalapril and captopril are considered. [e] ARBs: Angiotensin receptor blockers (ARBs) or angiotensin II receptor antagonists. Losartan potassium is considered. [f] Intravenous iron is considered for 50% of patients with stage 5, one time per month. [g] Erythropoietin is considered for 50% of patients with stage 5, two times a week.
(PDF)

**S4 Table. Treatment included for ESKD.** Adapted from data extracted from the Chilean Individual Expected Cost Verification Study (EVC), the Chilean National Health Fund and experts' opinion. HD: Haemodialysis. PD: peritoneodialysis. [a] Annual use per patient based on the Expected Cost Verification Study in 2019. [b] Frequency of use considered as the percentage of patients that would use the specific treatment. [c] For simplification, we included all the types of vascular access covered by Chilean healthcare system in this section. The election of it will depend on the need of the patient and decided by the specialist. [d] For simplification, we grouped all the laboratory tests considered for individuals in stage 5. [e] Intravenous iron is considered for all patients undergoing dialysis, around 3 times per month. [f] Erythropoietin is considered for all patients undergoing dialysis, two times a week for 9 months per year.
(PDF)

**S5 Table. Costs included for the treatment of DM[a].** Adapted from data extracted from the Chilean Individual Expected Cost Verification Study (EVC), the Chilean National Health Fund and experts' opinion. DM: Diabetes Mellitus. [a] These costs consider only the treatment for DM that is not covered by the treatment of CKD (e.g. physicians consultations were not considered in the costing as we assumed the ones used for each CKD stage). [b] Annual use per patient based on the Expected Cost Verification Study in 2019. [c] Frequency of use considered as the percentage of patients that would use the specific treatment.
(PDF)

## Author Contributions

**Conceptualization:** Magdalena Walbaum, Shaun Scholes, Rubén Rojas, Jennifer S. Mindell, Elena Pizzo.

**Formal analysis:** Magdalena Walbaum, Shaun Scholes, Rubén Rojas.

**Investigation:** Magdalena Walbaum.

**Methodology:** Magdalena Walbaum, Shaun Scholes, Rubén Rojas, Elena Pizzo.

**Software:** Magdalena Walbaum.

**Visualization:** Magdalena Walbaum, Rubén Rojas.

**Writing – original draft:** Magdalena Walbaum.

**Writing – review & editing:** Shaun Scholes, Rubén Rojas, Jennifer S. Mindell, Elena Pizzo.

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
