## [Decision Letter · Decision Letter 0]

17 Mar 2021

PONE-D-20-30503

Projection of the health and economic impacts of Chronic Kidney Disease in the Chilean population

PLOS ONE

Dear Dr. Pizzo,

Thank you for submitting your manuscript to PLOS ONE. After careful consideration, we feel that it has merit but does not fully meet PLOS ONE’s publication criteria as it currently stands. Therefore, we invite you to submit a revised version of the manuscript that addresses the points raised during the review process.

Please can you address Reviewer 1's comments particularly carefully, especially point 1.

We look forward to receiving your revised manuscript.

Kind regards,

Lucy C. Okell

Academic Editor

PLOS ONE

Journal Requirements:

Reviewers' comments:

Reviewer's Responses to Questions

**Comments to the Author**

1. Is the manuscript technically sound, and do the data support the conclusions?

Reviewer #1: Partly

Reviewer #2: Yes

2. Has the statistical analysis been performed appropriately and rigorously? 

Reviewer #1: No

Reviewer #2: Yes

3. Have the authors made all data underlying the findings in their manuscript fully available?

Reviewer #1: No

Reviewer #2: No

4. Is the manuscript presented in an intelligible fashion and written in standard English?

Reviewer #1: Yes

Reviewer #2: Yes

5. Review Comments to the Author

Reviewer #1: Thank-you for the opportunity to review this work. Population health health economic analyses are an important component of optimal planning in developing national strategies for CKD care. The intent of this analysis comes from the right place and the manuscript is well written. I do have some significant concerns with the assumptions underlying the population of the model.

1. The authors refer to the 2012 KDIGO CKD staging system in the set up of the model and establishing transition probabilities from state to state. The 2012 CKD staging system however is not purely GFR based as outlined in the manuscript. Urine albumin to creatinine ratios are a far more potent risk factor for CKD progression than GFR alone. To not consider uACR values in base case establishment and the impact on therapies to mitigate transitions such as BP control, ACE-ARB's and SGLT-2 inhibitors render this model obsolete in terms of what we now know of CKD transitions. Some comment or incorporation of uACR in the "heat map" format suggested in the 2012 guidelines are necessary to consider this model novel and relevant by today's standards.

2. As mentioned above, transition probabilities between states is significantly affected by therapies - ACE/ARB and SGLT-2 inhibitors in addition to BP control and now mineralocorticoid antagonists. A sensitivity analysis adding in population level prevalence of these therapies over time would strengthen the validity of the model.

3. While transplant, PD and Home HD are not currently utilized much in Chile, they are demonstrably cheaper and better therapies for ESRD in most cases. Sensitivity analysis of incorporating more of these would be helpful in projecting downstream costs and help payors and planners with their resource allocations moving forward.

4. More transparency on costing estimates are needed. 8273 GBP for HD care seems very low even if HR rates are lower in Chile, consumables and machine costs are much higher than this annually. If ESRD rates grow, capital costs of constructing more HD units are substantial and need to be considered as well. I'm not sure this has been done.

5. A discount rate of 3% has been applied. We would suggest adjusting for inflation using medical consumer price index for health care costs.

Reviewer #2: Thank you for the opportunity to reveiw this article. I found it to be an interesting and well-conducted analysis of CKD progression and costs in Chile. I have the following comments/suggestions

- I would appreciate more clarity around the regression-fitting described for the initial prevalence of CKD. Specifically, is this equation provided by teh ENS report, or calculated by the authors? If it's the latter, what were the specific data used for this estimation and was there evidence that the log-transformed linear equation well fit the distribution? Generally, if the ENS report provided number of individuals at eGFR levels it isn't clear to me where the need for regression comes in. It may well be a sound approach, but more details about the underlying data, statistical diagnostics, and use of the values within the model would help for transparency

- The increasing number of CKD cases overall in the population is a key model outcome, and more discussion is warranted on the underlying assumptions leading to this. e.g. based on the inputs, it would seem that in terms of total CKD cases growing in time, this is primarily due to population growth and aging population - thus, it would not be just CKD but other chronic diseases expected to increase in time, and general discussion of these trends is warranted

- Sensitivity analyses were conducted around the proportion of fast vs. slow progressors, but not for the progression rates within the respective categories. These are key parameters, particularly around eventual progression to RRT as a cost-driver. The authors provide citations to other publications around these values but further discussion of them (and potentially testing alternative values) would strengthen the analysis - e.g. does this rate of progression reflect current clinical experience in Chile? Might it change based on available treatments, etc.?

- The cost inputs described in the methods and Table 2 require additional detail - how were they calculated? Are the values in Table 2 based on published values by health state, or were they microcosted based on specific health resources required in each stage of disease (and if so, what inputs were used for this)? This level of detail may not be required within the main body of the paper, but inclusion in an appendix would improve the transparency of analysis

6. PLOS authors have the option to publish the peer review history of their article (what does this mean?). If published, this will include your full peer review and any attached files.

Reviewer #1: No

Reviewer #2: No

---

## [Author Response · Author response to Decision Letter 0]

2 Aug 2021

We thank the reviewers for their thoughtful comments. We have responded to each point below (indicated in bold) and have highlighted changes to the manuscript or appendix in blue.

Comments to the Author

1. Is the manuscript technically sound, and do the data support the conclusions?

Reviewer #1: Partly

Reviewer #2: Yes

We hope with the revised version we achieve a technically sound piece of scientific research.

2. Has the statistical analysis been performed appropriately and rigorously? 

Reviewer #1: No

Reviewer #2: Yes

We included more scenarios to take into account the structural uncertainty of the model. E.g. we included iSGLT2 for the treatment of Stages 3a and 3b and pre-dialysis for stages 4 and 5. We included in the supplementary material the equations used in the model.

3. Have the authors made all data underlying the findings in their manuscript fully available?

Reviewer #1: No

Reviewer #2: No

The raw data of the Chilean national health surveys are freely available upon request. We stated in the manuscript: Data Sharing Statement: The full data sets of the national health surveys (Encuesta Nacional de Salud ENS) can be accessed in through the Ministry of Health of Chile website found at: http://epi.minsal.cl/encuestas-poblacionales/

We also included in the supplementary material all the aggregated data underlying the findings of the manuscript.

Reviewer #1: 

Thank-you for the opportunity to review this work. Population health health economic analyses are an important component of optimal planning in developing national strategies for CKD care. The intent of this analysis comes from the right place and the manuscript is well written. I do have some significant concerns with the assumptions underlying the population of the model.

Thank you for your comments.

1. The authors refer to the 2012 KDIGO CKD staging system in the set up of the model and establishing transition probabilities from state to state. The 2012 CKD staging system however is not purely GFR based as outlined in the manuscript. Urine albumin to creatinine ratios are a far more potent risk factor for CKD progression than GFR alone. To not consider uACR values in base case establishment and the impact on therapies to mitigate transitions such as BP control, ACE-ARB's and SGLT-2 inhibitors render this model obsolete in terms of what we now know of CKD transitions. Some comment or incorporation of uACR in the "heat map" format suggested in the 2012 guidelines are necessary to consider this model novel and relevant by today's standards.

Thank you for this comment. We have clarified this point in the manuscript, with the inclusion of the uACR categories for the classification of the stages:

“The CKD stages were defined based on the Kidney Disease: Improving Global Outcomes (KDIGO) 2012 classification (Supplementary material Figure S1): Stage 3a: eGFR 45–59 ml/min/1.73m2 with normal or increased albuminuria; Stage 3b: eGFR 30–44 ml/min/1.73m2 with normal or increased albuminuria; Stage 4: eGFR 15–29 ml/min/1.73m2 with normal or increased albuminuria, and Stage 5: eGFR <15 ml/min/1.73m2 with normal or increased albuminuria.”

We also added the “heat map” in the supplementary materials and calculated the proportion of increased albuminuria by stage to determine the proportion of fast or slow progressors by stage. 

2. As mentioned above, transition probabilities between states is significantly affected by therapies - ACE/ARB and SGLT-2 inhibitors in addition to BP control and now mineralocorticoid antagonists. A sensitivity analysis adding in population level prevalence of these therapies over time would strengthen the validity of the model.

Thank you for this suggestion. We added the effect of SGLT2 inhibitors as part of a new analysis and present the results in Table 3 and 4

3. While transplant, PD and Home HD are not currently utilized much in Chile, they are demonstrably cheaper and better therapies for ESRD in most cases. Sensitivity analysis of incorporating more of these would be helpful in projecting downstream costs and help payors and planners with their resource allocations moving forward.

Thank you for this suggestion. We included in the sensitivity analysis a +-20% in the costs to account for the possibility of increasing the PD in Chile. We know that renal transplantation is the most cost-effective option for RRT, but due to unavailability of data we could not include it in the model. We included a brief discussion in the limitations of the model: “Also, due to unavailability of Chilean renal transplantation data we could not include this treatment in the model and limited our simulation and analyses of RRT only to dialysis.”.

4. More transparency on costing estimates are needed. 8273 GBP for HD care seems very low even if HR rates are lower in Chile, consumables and machine costs are much higher than this annually. If ESRD rates grow, capital costs of constructing more HD units are substantial and need to be considered as well. I'm not sure this has been done.

Thank you for the opportunity to clarify this point. We requested new data from the Ministry of health and added the tables with the quantities and costs considered for the model (supplementary material for the full data). 

5. A discount rate of 3% has been applied. We would suggest adjusting for inflation using medical consumer price index for health care costs.

Thank you for this suggestion. We adjusted for inflation using medical consumer price index.

Reviewer #2: Thank you for the opportunity to review this article. I found it to be an interesting and well-conducted analysis of CKD progression and costs in Chile. I have the following comments/suggestions

Thank you for your comments.

- I would appreciate more clarity around the regression-fitting described for the initial prevalence of CKD. Specifically, is this equation provided by the ENS report, or calculated by the authors? If it's the latter, what were the specific data used for this estimation and was there evidence that the log-transformed linear equation well fit the distribution? Generally, if the ENS report provided number of individuals at eGFR levels it isn't clear to me where the need for regression comes in. It may well be a sound approach, but more details about the underlying data, statistical diagnostics, and use of the values within the model would help for transparency

Thank you for the opportunity to clarify this point. We added to the manuscript:

“The fitted equation was ln(y) = -0.0556 + (0.077*x); predicted values were obtained by using the exponential transformation (Supplementary material, Figure S3). For example, the estimated number at eGFR=59ml/min/1.73m2 was equal to exp(-0.0556 + (0.077*59)) ≈90. Using the fitted equation of the discrete values of eGFR in the Chilean population, we could estimate the proportion of individuals in each stage that progressed at the end of each cycle based in the different progression rates assumed in the model (of 3 and 5 ml/min/1.73m2).”

- The increasing number of CKD cases overall in the population is a key model outcome, and more discussion is warranted on the underlying assumptions leading to this. e.g. based on the inputs, it would seem that in terms of total CKD cases growing in time, this is primarily due to population growth and aging population - thus, it would not be just CKD but other chronic diseases expected to increase in time, and general discussion of these trends is warranted

Thank you for your comment. We do not have data on Chilean incidence of CKD so we had to assume the incidence given by the literature, adjusted by the adult population growth. We included the incidence of CKD in people with and without diabetes given by the literature.

We have discussed the trends in the increase in risk factors in the manuscript: “Other risk factors behind the rise in CKD prevalence are probably the increase in the prevalence of diabetes, hypertension, and obesity in the general population. In Chile, according to the last health survey (2016-17), 12.3% of adults had survey-defined diabetes, 34.6% were obese and 27.7% had survey-defined hypertension. These show a significant increase in diabetes and obesity compared with the previous health survey in 2009-10, with prevalence of 9.4% and 25.7% respectively.”

- Sensitivity analyses were conducted around the proportion of fast vs. slow progressors, but not for the progression rates within the respective categories. These are key parameters, particularly around eventual progression to RRT as a cost-driver. The authors provide citations to other publications around these values but further discussion of them (and potentially testing alternative values) would strengthen the analysis - e.g. does this rate of progression reflect current clinical experience in Chile? Might it change based on available treatments, etc.?

Thank you for this comment. We calculated the proportion of increased albuminuria by stage and then determined the proportion of fast, medium or slow progression by stage depending on the proportion of individuals with increased albuminuria+ diabetes (fast progression), diabetes alone (medium progression) or no diabetes and no increased albuminuria (slow progression). We included the proportion of individuals with increased albuminuria by CKD stage in Table 1. We re-did all the analyses with these changes and show the equations in the supplementary material.

- The cost inputs described in the methods and Table 2 require additional detail - how were they calculated? Are the values in Table 2 based on published values by health state, or were they microcosted based on specific health resources required in each stage of disease (and if so, what inputs were used for this)? This level of detail may not be required within the main body of the paper, but inclusion in an appendix would improve the transparency of analysis

Thank you for the opportunity to clarify this point. We added the tables with the quantities and costs considered for the model (see supplementary material).

---

## [Editor Report · Decision Letter 1]

13 Aug 2021

Projection of the health and economic impacts of Chronic Kidney Disease in the Chilean population

PONE-D-20-30503R1

Dear Dr. Pizzo,

We’re pleased to inform you that your manuscript has been judged scientifically suitable for publication and will be formally accepted for publication once it meets all outstanding technical requirements.

Kind regards,

Lucy C. Okell

Academic Editor

PLOS ONE
---

## [Editor Report · Acceptance letter]

20 Aug 2021

PONE-D-20-30503R1 

Projection of the health and economic impacts of Chronic Kidney Disease in the Chilean population. 

Dear Dr. Pizzo:

I'm pleased to inform you that your manuscript has been deemed suitable for publication in PLOS ONE. Congratulations! Your manuscript is now with our production department. 

Kind regards, 

on behalf of

Dr. Lucy C. Okell 

Academic Editor

PLOS ONE